# The Role of Somatostatin Analogues in the Control of Diarrhea and Flushing as Markers of Carcinoid Syndrome: A Systematic Review and Meta-Analysis

**DOI:** 10.3390/jpm13020304

**Published:** 2023-02-09

**Authors:** Krystallenia I. Alexandraki, Anna Angelousi, Eleftherios Chatzellis, Alexandra Chrisoulidou, Nikolaos Kalogeris, Georgios Kanakis, Christos Savvidis, Dimitra Vassiliadi, Ariadni Spyroglou, Georgios Kostopoulos, Vyron Markussis, Konstantinos Toulis, Stylianos Tsagarakis, Gregory A. Kaltsas

**Affiliations:** 1Endocrine Unit, 2nd Department of Surgery, Aretaieio Hospital, National and Kapodistrian University of Athens, 11528 Athens, Greece; 2Unit of Endocrinology, 1st Department of Internal Medicine, Laiko University Hospital, Medical School, National and Kapodistrian University of Athens, 11527 Athens, Greece; 3251 HAF and VA Hospital, 11525 Athens, Greece; 4Unit of Endocrinology, Theagenio Hospital, 54639 Thessaloniki, Greece; 5Department of Endocrinology and Diabetes, Hellenic Red Cross Hospital, 11526 Athens, Greece; 6Department of Endocrinology, Diabetes and Metabolism, Athens Naval & VA Hospital, 11521 Athens, Greece; 7Department of Endocrinology and Metabolism, Hippocratio General Hospital of Athens, 11527 Athens, Greece; 8Department of Endocrinology, Diabetes and Metabolism, Evangelismos Hospital, 10676 Athens, Greece; 9Department of Endocrinology, 424 General Military Hospital, 56429 Thessaloniki, Greece; 10Independent Researcher, 18535 Athens, Greece; 11Endocrine Unit, 1st Department of Propaedeutic Medicine, Laiko University Hospital, Medical School, National and Kapodistrian University of Athens, 11527 Athens, Greece

**Keywords:** somatostatin analogues, neuroendocrine tumors, carcinoid syndrome

## Abstract

Background: Somatostatin analogues (SSAs) are the cornerstone of treatment for carcinoid syndrome (CS)-related symptoms. The aim of this systematic review and meta-analysis is to evaluate the percentage of patients achieving partial (PR) or complete response (CR) with the use of long-acting SSAs in patients with CS. Methods: A systematic electronic literature search was conducted in PubMed, Cochrane, and Scopus to identify eligible studies. Any clinical trials reporting data on the efficacy of SSAs to alleviate symptoms in adult patients were considered as potentially eligible. Results: A total of 17 studies reported extractable outcomes (PR/CR) for quantitative synthesis. The pooled percentage of patients with PR/CR for diarrhea was estimated to be 0.67 (95% confidence interval (CI): 0.52–0.79, I_2_ = 83%). Subgroup analyses of specific drugs provided no evidence of a differential response. With regards to flushing, the pooled percentage of patients with PR/CR was estimated to be 0.68 (95% CI: 0.52–0.81, I_2_ = 86%). Similarly, no evidence of a significant differential response in flushing control was documented. Conclusions: We estimate there is a 67–68% overall reduction in symptoms of CS associated with SSA treatment. However, significant heterogeneity was detected, possibly revealing differences in the disease course, in management and in outcome definition.

## 1. Introduction

Neuroendocrine tumors (NETs) represent a heterogeneous group of neoplasms arising from neuroendocrine cells of the endocrine system [1]. They can develop in almost any organ, but occur most frequently in the lung, pancreas, and gastrointestinal tract [2]. Although the majority of these tumors are non-functioning, approximately 5–20% of NETs, depending also on the origin of the tumor, present hormonal hypersecretion, with carcinoid syndrome (CS) being the most frequent [3,4,5]. CS is clinically defined by diarrhea and flushing [6], sometimes accompanied by abdominal pain, facial telangiectasias, wheezing and tachycardia [7]. Diarrhea is caused by serotonin oversecretion, but CS symptoms are also attributed to histamine, kallikrein, prostaglandins, tachykinins, substance P, and other bioactive substances. Biochemically, CS is defined by elevated 24-h urine 5-hydroxyindolacetic acid (5-HIAA), the main metabolite of serotonin; however, serum 5-HIAA and plasma/platelet and spot urine serotonin measurements, possibly exhibiting a higher sensitivity, are not widely available yet [8]. Metastatic midgut NETs are the most common causes of CS and can also be encountered in patients with lung carcinoids, extensive mesenteric disease, ovarian NETs and rarely pancreatic NETs [2,9]. CS has a major effect on patients’ well-being, worsens the quality of life and is related to worse overall survival [3,6,10]. A major long-term and devastating complication of patients with advanced NETs and CS is the development of carcinoid heart disease (CHD), due to fibrotic plaques in the right heart, resulting in tricuspid valve regurgitation and leading to right heart failure. CHD development and subsequent progression correlates with high 5-HIAA, particularly at levels > 300 μmol/24 h [6,7,11]. Surgical resection is the treatment of choice for local, locoregional disease and oligometastatic hepatic disease and as a means for cytoreduction in severe and uncontrolled CS [12,13]. However, somatostatin analogues (SSAs) remain the standard medical care for the hormonal symptoms in CS, even in a presurgical setting, in order to avoid a carcinoid crisis [14,15]. Somatostatin (SST-14), originally discovered in 1973, was considered at the time as a hypothalamic neuropeptide inhibiting the secretion of growth hormone but subsequently was shown to also inhibit gastrointestinal cell motility, secretion, and growth [16]. In turn, already in 1978, endogenous somatostatin was found to alleviate flushing and diarrhea in patients with CS but its action was hampered due to its short half-life necessitating a twice- or thrice dosing schedule [16,17]. Subsequently, long-acting SSAs were developed to ensure sustained drug levels [18,19]. At present, according to the European Neuroendocrine Tumor Society (ENETS) and the European Society for Medical Oncology (ESMO) guidelines, the following SSA schemes are recommended for the symptom control of CS: first generation SSAs, octreotide long-acting repeatable (LAR) (30 mg), injected intramuscularly once every 4 weeks; and lanreotide AutoGel (AG) (120 mg) injected subcutaneously once every four weeks, with shortening of the injection interval to 3 or even 2 weeks to achieve symptom control. Second generation SSA, pasireotide LAR, may be considered as an off-label treatment in cases where other SSAs fail to control symptoms [6,20]. According to the ENETS guidelines for the perioperative prophylaxis of a carcinoid crisis, patients pre-treated with long-acting SSAs should continue their treatment, and subcutaneous administration of octreotide at 100–200 µg 2–3 times/day should occur for 2 weeks prior to surgery, whereas acute intravenous octreotide should also be readily available intraoperatively [11]. SSA administration does not only correlate with significant symptom control in CS but also with reduction in urinary levels of 5-HIAA, the most accurate biomarker of CS. However, in the majority of the relevant studies, the biochemical response appears less prominent than the clinical response [6,21,22].

Octreotide and lanreotide have a high affinity for the somatostatin receptor type 2 (SSTR2) and a modest affinity for SSTR5 [23]. Pasireotide LAR is a novel SSA with high affinity to SSTR 1-3 and 5 [24,25,26]. Most NETs express multiple SSTRs, rendering all three SSAs possibly effective in CS [27]. However, variations in efficacy, and more specifically the escape phenomenon observed after an initial symptom control, can be attributed to desensitization due to SSTR internalization in tumor cells or alterations in the equilibrium of the various SSTR types [26,28]. Refractory CS, with loss of symptom control under the established SSA dosage occurs, though, most commonly due to tumor progression [29]. The first treatment options in refractory CS are either a dose escalation of SSAs, the shortening of the interval between SSA administration, or the additional administration of short-acting SSAs [30]. Three systematic reviews have concluded that high-dose SSAs are well-tolerated and achieve significant rates of disease control [31,32,33]. In patients with CS symptoms refractory to octreotide LAR or lanreotide AG, pasireotide LAR showed non-inferiority in symptom control in comparison with high dose (40 mg) octreotide LAR [34]. Furthermore, in patients with refractory CS, additional therapies inhibiting serotonin synthesis (telotristat ethyl), along with treatments to reduce tumor burden, such as locoregional therapies and embolization, and systemic treatments, including peptide receptor radionuclide therapy (PRRT), everolimus and interferon-α, are used in addition to SSAs to achieve tumor control and improve CS symptoms. The “Telotristat Etiprate for Somatostatin Analogue Not Adequately Controlled Carcinoid Syndrome” (TELESTAR) and the “Telotristat Etiprate for Carcinoid Syndrome Therapy” (TELECAST) trials have demonstrated the efficacy of telotristat ethyl either alone or as add-on to SSAs in symptom control in patients with CS-related diarrhea, together with a significant biochemical response with a significant reduction in the 5-HIAA levels [35,36,37]. PRRT with ^177^Lutetium-DOTATATE has also been proven to be effective in patients with refractory CS, achieving a significant decrease in diarrhea and flushing. In the “Study Comparing Treatment with 177-Lu-DOTA0-Tyr3-Octreotate to Octreotide LAR in Patients with Inoperable, Progressive, Somatostatin Receptor Positive Midgut Carcinoid Tumours” (NETTER-1) significant symptom control was reported in the ^177^Lutetium-DOTATATE group compared with the control group that was treated with above-label doses of SSA [6,38,39]. 

There is a well-established safety profile for SSAs, attributable to their clinical use in the treatment for NETs for over two decades. SSAs are usually well-tolerated, and the most frequent adverse events (AEs) include abdominal pain with cramps, constipation, diarrhea, steatorrhea, injection-site irritation, nausea, and vomiting [40,41,42]. Impaired glucose tolerance or even overt diabetes mellitus has been seen in selected cases of patients administered long-acting SSAs [43,44].

Although SSAs are the mainstay of treatment for CS, controversy still exists regarding the extent and duration of their efficacy in controlling CS symptoms. Two meta-analyses have so far reviewed the efficacy of SSAs in symptomatic control of carcinoid syndrome [41,45]. The purpose of this study was to perform a systematic review of the literature and meta-analysis in order to evaluate the percentage of patients with CS achieving partial (PR) or complete response (CR) regarding diarrhea and flushing episodes upon treatment with the long-acting formulations of SSAs (lanreotide AG, octreotide LAR, and pasireotide LAR) that are currently used in everyday clinical practice. 

## 2. Materials and Methods

### 2.1. Search Strategy

For the present systematic review and meta-analysis, PubMed, Cochrane databases and Scopus were searched for relevant English language studies for inclusion. The search included studies published from inception of the respective database until September 2022. This review was conducted according to the Preferred Reporting Items for Systematic Reviews and Meta-Analyses (PRISMA) guidelines (PROSPERO ID: CRD42023387392). The search keywords included: somatostatin analogues (lanreotide, octreotide, pasireotide), carcinoid syndrome/heart disease/tumor, neuroendocrine tumor/carcinoma, and gastro-enteropancreatic neuroendocrine tumor. An ancillary search in the references of the retrieved studies complemented the search strategy. Nine reviewers (KIA, AA, EC, AC, NK, GK, CS, DV, and AS) independently screened the titles and abstracts of all the studies fulfilling the inclusion criteria and eligible full texts were then screened in detail by all the reviewers. In cases of disagreement, an additional reviewer (KT) determined eligibility. The articles were critically appraised to assess the risk of bias. An overall judgement of the methodological quality of the eligible studies was based on the following criteria: selection of participants, data on exposure and outcome, and reporting, whereas non-human studies, reviews, non-English studies and studies lacking data on the clinical response of diarrhea and/or flushing during SSA treatment were excluded. 

Any clinical trial reporting the major endpoint (the rate of improvement (PR, CR) in frequency or/and intensity of diarrhea or/and flushing), reporting data on the efficacy of SSAs, including at least 7 adult participants, using of current long-releasing formulations, and having at least three months minimum follow-up was considered as potentially eligible for the meta-analysis. A subset of studies, in which outcomes were reported as partial response (PR) or complete response (CR), was considered for meta-analysis of binomial data. Due to a lack of uniformity and established criteria for defining the response to treatment, definitions of PR and CR (variably defined in different studies) were based on ad hoc criteria provided in individual studies. In brief, some studies defined CR as the disappearance of symptoms and PR as the improvement in symptom frequency, while others defined CR as a >50% (or in one study >20%) reduction in the frequency of the symptoms. Some studies focused on the statistically significant reduction in the number of episodes upon treatment, without defining CR and PR separately. Thus, in our meta-analysis, CR and PR results of each separate study were grouped.

For each study included in the present analysis, the first author, publication year, study characteristics, and patient demographics were registered. The extracted data included: the number of enrolled patients, diagnosis, drug, dosage and frequency, duration of treatment, overall efficacy (PR and CR), efficacy on diarrhea (PR and CR), and efficacy on flushing (PR and CR). 

### 2.2. Meta-Analysis

We performed a random effects proportion meta-analysis using the inverse variance method with logit transformation for evidence synthesis of the primary outcomes (diarrhea and flushing). We calculated the 95% CI using the Clopper–Pearson method and estimated the between-study variance (heterogeneity) with I^2^ and τ^2^. Heterogeneity was considered to be low if I² = 25%, moderate if I² = 50%, or high if I² = 75%. We also conducted prespecified subgroup analyses to investigate and compare the treatment effects of the different long-acting SSAs (lanreotide, octreotide LAR, and pasireotide) on the outcomes of interest. All analyses were performed using R Project for Statistical Computing (version 3.6.3). Specifically, we used the “metaprop” function from the “meta” package to conduct the primary meta-analyses and additional subgroup analyses.

## 3. Results

### 3.1. Descriptive Characteristics of Included Studies

A total of 2913 articles were identified through the literature search, and after screening by title and abstract, 59 articles remained for full-text review (Figure 1). Finally, 17 articles, published between 1994 and 2022, with a total of 970 patients, met the eligibility criteria for our research question, had extractable outcomes (PR or CR) and were included in our systematic review and meta-analysis. Out of these, nine studies investigated the effect of Lanreotide AG 120 mg sc or 30 mg im (322 patients), eight studies investigated the effect of octreotide LAR 10–40 mg (605 patients) and one study concerned the effect of pasireotide 20–60 mg LAR (43 patients); this study compared the effects of octreotide LAR with pasireotide [21,22,34,46,47,48,49,50,51,52,53,54,55,56,57,58,59]. 

### 3.2. Systematic Review 

#### 3.2.1. Octreotide

In a large retrospective study, treatment with octreotide LAR at varying doses (20 or 30 or 40 mg every 28 days) led to symptom resolution or improvement in 60% of patients with flushing and in 48% of patients suffering from diarrhea. Furthermore, after 12 months of octreotide LAR administration, 48% and 30% of treated patients showed improvement in or resolution of bronchoconstriction and improvement in CHD, respectively. The most common AEs reported in this study were hyperglycemia, cholelithiasis, cholecystitis and steatorrhea [55]. In another small study, octreotide LAR at a dosage of 20 mg per month led to a disappearance of diarrhea in 9 out of 10 patients and of flushing episodes in 7 out of 10 patients, while the remaining patients had significant improvement in their symptoms. Additionally, in CS patients with high baseline 5-HIAA levels, octreotide LAR administration led to a significant reduction in this metabolite [47]. Among the CS patients included in the “Placebo-controlled, double-blind, prospective, randomized study on the effect of octreotide LAR in the control of tumor growth in patients with metastatic neuroendocrine midgut tumors” (PROMID), 70% presented PR or CR for flushing after 6 months of treatment with 30 mg octreotide LAR every 28 days, vs. 25% in the placebo arm. The most frequent AEs observed were related to the gastrointestinal tract and the hematopoietic system [46]. In a multicenter open-label preliminary study, the administration of octreotide LAR every 21 days upon disease progression led to a CR of 60% and PR of 40% in patients with endocrine symptoms (diarrhea or flushing), without causing additional toxicity, with most AEs being diarrhea, abdominal pain, cholelithiasis and pyrexia [48]. A multicenter retrospective study demonstrated an improvement in flushing by 81% and in diarrhea by 79% among patients receiving octreotide LAR above 30 mg every 4 weeks [49]. In a small prospective study investigating the effect of 30 mg octreotide LAR every 28 days in therapy-naïve CS patients, the drug resulted in a significant reduction in bowel movements and a reduction in diarrhea in 11 out of 12 included patients, together with a biochemical response corresponding to a significant decrease in 5-HIAA [53]. The control arm of a randomized phase III trial aiming at comparing pasireotide LAR with octreotide LAR (40 mg every 4 weeks) in patients with refractory CS upon maximum approved SSA dosage showed symptom control of 27% in the octreotide LAR arm, while no drug related events leading to drug discontinuation were documented in this arm [34].

#### 3.2.2. Lanreotide

In a retrospective study, administration of lanreotide depot at a dosage of 60 mg every 28 days in 23 patients, 90 mg every 28 days in 36 patients or 120 mg every 28 days in 7 patients led to a clinical response in 95% of the patients with diarrhea and in 91% of patients with flushing, resulting in an in-total 94% symptomatic response for both symptoms at follow-up, with steatorrhea and cholelithiasis being the most common AEs [57]. In a monocentric study, the fortnightly administration of lanreotide depot led to a reduction in flushing in 86% and of diarrhea in 42% of CS patients. The AEs included injection site disturbances and cholelithiasis [50]. Ricci et al. described an overall symptomatic response of 70% in patients receiving fortnightly lanreotide depot injections [52]. In an open multicenter crossover study comparing lanreotide 30 mg every 10 days versus octreotide 200 µg twice or thrice daily, an improvement in flushing was documented in 54% of lanreotide patients (vs. 68% in the octreotide group) and in diarrhea in 45% of the lanreotide patients (vs. 50% in the octreotide group) [22]. A small study, evaluating the administration of lanreotide depot at a dose of 30 mg every 10 days in CS patients documented a complete resolution of diarrhea and flushing in all included symptomatic patients. The therapy was well tolerated, with transient pain at the injection site being the only AE registered herein [51]. In a prospective multicenter study, the fortnightly administration of lanreotide depot achieved a significant reduction in diarrhea (56%) and flushing (54%) episodes in CS patients, with a significant reduction in 5-HIAA levels in 42% of patients. The AEs included pain or redness at the injection site and cholelithiasis [59]. Ruszniewski et al. subsequently investigated the efficacy of lanreotide AG (90 mg every 4 weeks for first 2 months, then dose titration to 60 mg or 120 mg) for diarrhea and flushing in patients with CS and documented a 65% improvement in diarrhea and an 18% improvement in flushing, with diarrhea (11%) and injection site disorders (4%) being the most frequent AEs, leading to discontinuation of treatment in 3% of included patients [21]. In a multicenter, open-label, phase II trial conducted in Spain that examined the administration of lanreotide AG 120 mg every 28 days, 56% of the CS patients experienced a complete resolution of diarrhea, while asthenia, flatulence and injection site pain were the most common AEs registered [54]. In the recent phase IV, prospective, open-label, exploratory study “Circulating Tumour Cells in Somatuline Autogel Treated NeuroEndocrine Tumors Patients” (CALM-NET) investigating the effect of the administration of 120 mg lanreotide AG every 28 days on CS symptoms, 70.6% of treated patients had an at least 50% reduction in diarrhea episodes and 77.1% in flushing episodes at follow-up [58]. 

#### 3.2.3. Pasireotide

A unique study investigating the efficacy of pasireotide LAR in regard to symptom control is the previously mentioned randomized phase III trial comparing pasireotide LAR to octreotide LAR (40 mg every 4 weeks). Treatment with pasireotide LAR, in patients with refractory CS while already receiving maximum approved doses of other SSAs led to a reduction in symptoms by 21%, showing non-inferiority in comparison with octreotide LAR regarding symptom control in CS (*p* = 0.53). This is notable given the higher than usual octreotide LAR dosage used in this study. In this arm, common AEs included hyperglycemia (28%), fatigue (11%) and nausea (9%), and 17% of the patients reported a serious AE (SAE), leading to discontinuation of the drug in 8% of the patients [34]. 

### 3.3. Primary Meta-Analysis

#### 3.3.1. Symptom: Diarrhea

The pooled percentage of patients with PR or CR for diarrhea was estimated to be 0.67 (16 studies, 95% confidence interval (CI): 0.52–0.79, I2 test for heterogeneity = 83%). Subgroup analyses on the basis of specific drugs provided no evidence of differential responses. For lanreotide and octreotide, the pooled percentage of patients with PR or CR was calculated to be 0.67 (95 CI%: 0.51–0.80) and 0.75 (95 CI%: 0.51–0.89), respectively. The relative paucity of data precluded the calculation of pasireotide-specific outcomes (Figure 2). When excluding the pasireotide data, the overall percentage of patients with PR or CR of diarrhea under octreotide LAR or lanreotide AG increased to 0.70 (95 CI%: 0.57–0.80) (Figure 3). 

#### 3.3.2. Symptom: Flushing 

The pooled percentage of patients with PR or CR for flushing was estimated to be 0.68 (12 studies 95% CI: 0.52–0.81, I2 = 86%). Similarly, no evidence of a significant differential response in the control of flushing was noted (for lanreotide, the pooled percentage of patients with PR or CR was 0.75 (0.58–0.86) and for octreotide it was 0.70 (0.46–0.87)). Here again, the relative scarcity of data prevented the calculation of pasireotide-specific outcomes (Figure 4). When excluding the pasireotide data, the overall percentage of patients with PR or CR for flushing under octreotide LAR or lanreotide AG increased to 0.72 (95 CI%: 0.59–0.82) (Figure 5).

## 4. Discussion

The systematic review and meta-analysis provided an estimated 67–68% overall reduction in symptoms of CS under SSA treatment. However, this finding was associated with significant heterogeneity among the included studies, potentially reflecting differences in the clinical course of the disease as well as in outcome definitions. In particular, the included studies presented significant heterogeneity in their design and follow-up, in the definition of the primary outcome (improvement vs. non-improvement), in the specific drug formulation and total dose administration of the investigated drugs, in the location of the primary tumor and extent of the disease. The exclusion of studies with pasireotide LAR did not significantly alter the level of heterogeneity. Additionally, there was no common mean (quantitative or qualitative) in assessing the response of diarrhea and flushing in retrospective studies. The absence of a predefined assessment tool of symptom response is also a potential limitation of the study. On the other hand, the overall symptom control estimated herein is in line with a previous meta-analysis performed in 2019, which included all SSA therapies available, that is, also short-acting SSAs, and demonstrated an overall control of the most common CS symptoms achieved in 66–70% of patients [45]. In our study, however, we included further recent available studies and focused only on the long-acting preparations (octreotide LAR, lanreotide AG, and pasireotide), which are mainly in use to date. In our study, when pasireotide treatment was excluded, the rates of symptoms reduction were slightly increased to 70–72%. Similarly, Modlin et al. demonstrated symptomatic relief in 74.2% of patients treated with octreotide LAR and in 67% of patients receiving lanreotide [41]. 

According to the vast majority of the previously published control studies, long-acting SSAs provide significant improvements in the CS symptoms, but no specific SSA drug has demonstrated superiority in symptom control. Similarly, our meta-analysis confirmed the efficacy of long-acting SSAs in CS symptom control, but the direct comparison of the pooled percentages of PR or CR between octreotide LAR and lanreotide AG did not provide any statistically significant difference between the two substances, neither in diarrhea, nor in flushing control.

In the literature, a direct comparison between two different long-acting SSAs has only been performed in a phase III study comparing pasireotide LAR with high-dose octreotide LAR [34], where both drugs showed similar efficacy in symptom control after 6 months of treatment. This was, practically, the only study with pasireotide LAR which could be included in our meta-analysis, thus preventing further conclusions on the efficacy of this substance in comparison with the other SSAs. Interestingly, though, when compared with the rest of the studies included in the meta-analysis, the disease control was significantly lower in both arms of this study, possibly suggesting methodological peculiarities in the design of the study. 

From the largest studies reporting AEs, hyperglycemia, cholelithiasis, injection site disturbances such as local pain, flatulence, steatorrhea and abdominal pain were reported for octreotide LAR [55,60]. However, no patient was reported to discontinue the medication due to the AEs. Gastrointestinal disorders, cholelithiasis and injection site disorders were reported for lanreotide AG, also leading to a very small percentage of dropouts (3%) [21]. No SAEs were reported for either drug, rendering them well-tolerated therapeutic options for symptom control in CS. In contrast, pasireotide LAR was associated with a rather large percentage of hyperglycemia (28%) and a similarly high percentage of SAEs (17%), leading to drug discontinuation in 8% of the patients [34]. However, due to its binding affinity for multiple SSTRs, pasireotide LAR is a possible rescue strategy in refractory CS. Overall, SSAs demonstrate a good safety profile compared with other therapies used for NET management [61]. In support of this notion, the “Study of Lanreotide Autogel in Non-functioning Entero-pancreatic Endocrine Tumors” (CLARINET) open-label extension (OLE) study showed that the overall incidences of AEs and treatment-related AEs were lower in the OLE than in the core study, without clinically meaningful differences in the incidences and type of serious or severe AEs between the two phases [62,63].

The SSAs octreotide LAR and lanreotide AG are the mainstay of treatment of CS symptoms and show no significant differences in their effectiveness, as the present and previous meta-analyses documented. Differential diagnosis in cases of deterioration of diarrhea during SSA therapy is, though, very important, and should include steatorrhea due to pancreatic insufficiency, bacterial overgrowth, malabsorption, or short bowel syndrome [6]. These causes should be excluded before considering therapy adjustment due to refractory CS. Dose escalation of SSAs to above-label doses in refractory CS, although still unapproved, has proven efficacy in symptom relief in several studies and is encouraged by the present guidelines, taking into account the AE profile of these drugs [6]. Two phase III studies investigated octreotide LAR in doses of 40 and 60 mg every 28 days and reported that, while these dosages are effective in controlling CS symptoms, they were not associated with a significant AE profile, with cholelithiasis and hyperglycemia being the most frequent events, occurring in all cases in less than 10% of patients [34,38]. Similarly, a phase II study reported the AE profile of lanreotide AG 120 mg every 14 days, which mainly included gastrointestinal disorders (two patients with hyperglycemia and single cases with cholelithiasis and cholecystitis) [64]. Furthermore, in refractory CS, SSA administration should never be discontinued, but for symptom control, additional therapies with telotristat ethyl as an add-on and/or PRRT should be initiated. Interferon-a is another option as an add-on to refractory CS treatment for patients on SSAs but can also be administered as first-line treatment in SSTR-negative patients [65]. For PRRT treatment and other local invasive treatments, short-acting SSAs should be used on top of the established long-acting SSA therapy to avoid a carcinoid crisis [6,11]. The regiment suggested includes intravenous administered octreotide at a starting dose of 50–100 μg/h with a mean dose of 100–200 μg/h, starting 12 h prior to the intervention, dose escalation to achieve symptom control and continuation for at least 48 h post-intervention [6,11]. SSAs play an additional pivotal role in CHD treatment in order to reduce 5-HIAA levels, along with conventional medical therapy including diuretics, mineralocorticoid receptor antagonists and cautious fluid replacement. Continuous SSA administration may be needed during surgical procedures, such as percutaneous or open valve replacement, to avoid a worsening of right cardiac failure or a carcinoid crisis in uncontrolled CS in these patients [6]. 

Our study has some limitations, the most critical being its heterogeneity. However, this meta-analysis offers an additional tool to confirm the value of previous meta-analyses which included short-acting formulations in addition to the long-acting ones that are currently used not only in everyday practice but also in a number of novel clinical trials as reported above.

In summary, this systematic review and meta-analysis highlights the overall efficacy associated with SSA administration in the symptom control of patients with CS. However, the significant heterogeneity of the included studies possibly suggests differences in the clinical manifestation and course of the disease, in outcome definition and in management in each center. Thus, it underlines the need for further prospective studies. As an example, a head-to-head comparison of the available therapeutic options with SSAs that stratifies the patients according to their tumor characteristics could further elucidate SSA efficacy in CS.

## Figures and Tables

**Figure 1 jpm-13-00304-f001:**
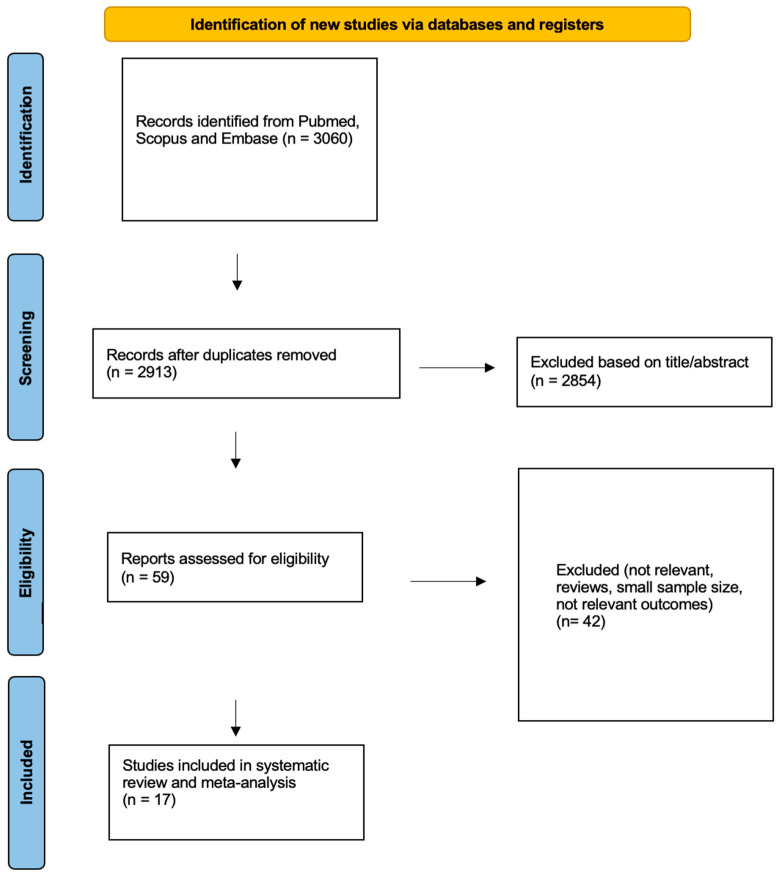
Flowchart for data collection according to PRISMA guidelines.

**Figure 2 jpm-13-00304-f002:**
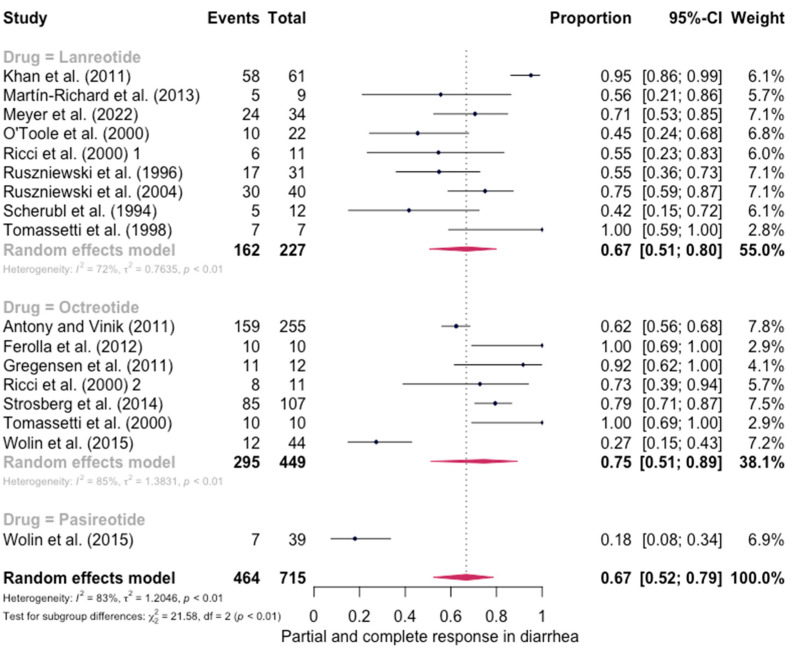
Forest plot showing the pooled PR and CR for diarrhea in patients with carcinoid syndrome. CI: confidence interval. p refers to Cochran’s Q test for heterogeneity [21,22,34,47,48,49,50,51,52,53,54,55,56,57,58,59].

**Figure 3 jpm-13-00304-f003:**
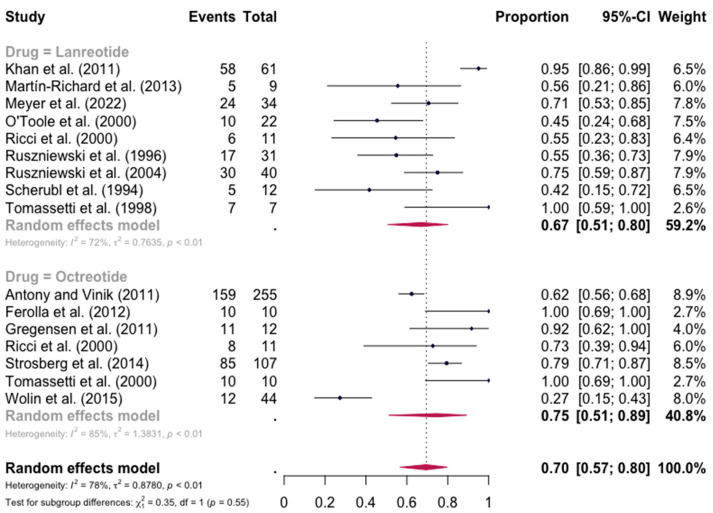
Forest plot showing the pooled PR and CR for diarrhea in patients with carcinoid syndrome, after excluding patients under pasireotide treatment. CI: confidence interval. p refers to Cochran’s Q test for heterogeneity [21,22,34,47,48,49,50,51,52,53,54,55,56,57,58,59].

**Figure 4 jpm-13-00304-f004:**
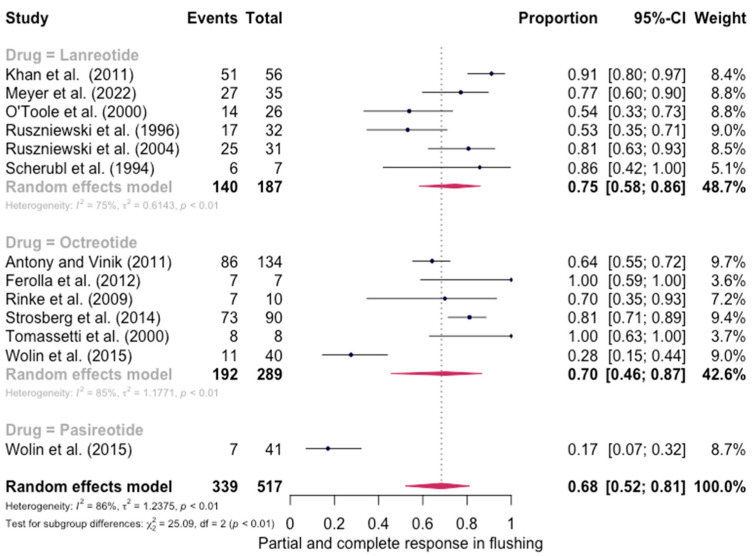
Forest plot demonstrating the pooled PR and CR for flushing in patients with carcinoid syndrome. CI: confidence interval, p: refers to Cochran’s Q test for heterogeneity [21,22,34,46,47,48,49,50,55,57,58,59].

**Figure 5 jpm-13-00304-f005:**
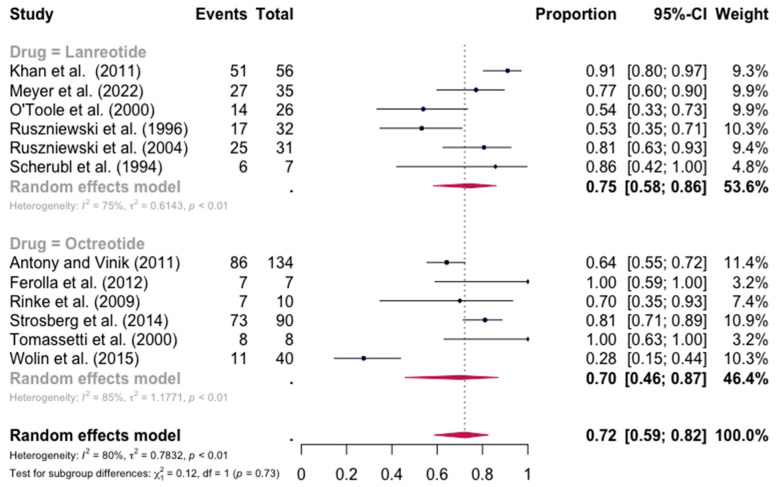
Forest plot demonstrating the pooled PR and CR for flushing in patients with carcinoid syndrome, after excluding patients under pasireotide treatment. CI: confidence interval, p: refers to Cochran’s Q test for heterogeneity [21,22,34,46,47,48,49,50,55,57,58,59].

## Data Availability

No new data were created or analyzed in this study. Data sharing is not applicable to this article.

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
