# Peer review of "The Role of Somatostatin Analogues in the Control of Diarrhea and Flushing as Markers of Carcinoid Syndrome: A Systematic Review and Meta-Analysis"

_jpm, 2023, doi:10.3390/jpm13020304_

Round 1
Reviewer 1 Report
The paper is well-written and focused on a quite interesting topic. However, some revisions are needed.
1. I'd suggest the authors to shorten the introduction, by removing the paragraph re. to the anti-proliferative effect of SSA as this is not the purpose of the review.
2. Introduction. I'd suggest to add some pertinent references: re. the paragraph on CS please add Spada F et al. Pharmaceuticals 2021 Jun 4;14(6):539; regarding the description of SSA in functioning tumors please add Massironi S et al. Scand J Gastroenterol. 2016;51(5):513-23
3. Introduction: when referring to refractory CS please add some brief data re. Telotristat
4. Could you please provide some data re. the effect of SSAs on 5HIAA levels, as it is the most accurate biomarker of CS?
5. Any data regarding the effect of high-dose/uncoventional SSAs on symptoms control?
6. Please carefully double check the paper for some grammar (i.e. this..provides, please add s, see discussion first line) and typo mistakes
Author Response
Review Report #1
The paper is well-written and focused on a quite interesting topic. However, some revisions are needed.
We would like to thank the reviewer for his helpful critique.
- I'd suggest the authors to shorten the introduction, by removing the paragraph re. to the anti-proliferative effect of SSA as this is not the purpose of the review.
According to the reviewer’s suggestion, this paragraph has now been removed from the revised version of the manuscript. In addition, we have adjusted the length of the introduction according to the comments of the Editor.
- Introduction. I'd suggest to add some pertinent references: re. the paragraph on CS please add Spada F et al. Pharmaceuticals 2021 Jun 4;14(6):539; regarding the description of SSA in functioning tumors please add Massironi S et al. Scand J Gastroenterol. 2016;51(5):513-23
As suggested by the reviewer, these references have now been added in the revised version of the manuscript (lines 58, 71 and 143).
- Introduction: when referring to refractory CS please add some brief data re. Telotristat
We would like to thank the reviewer for this highly relevant suggestion. We have now accordingly added the following text in the revised version of the manuscript.
“The “Telotristat Etiprate for Somatostatin Analogue Not Adequately Controlled Carcinoid Syndrome” (TELESTAR) and the “Telotristat Etiprate for Carcinoid Syndrome Therapy” (TELECAST) trials have demonstrated the efficacy of telotristat ethyl either alone or as add-on to SSAs in symptom control in patients with CS-related diarrhea, together with a significant biochemical response with a significant reduction in the 5-HIAA levels” (lines 118-123)
- Could you please provide some data re. the effect of SSAs on 5HIAA levels, as it is the most accurate biomarker of CS?
Data on the effect of SSAs on 5-HIAA have been added as follows:
“SSA administration does not only correlate with significant symptom control in CS but also with reduction in urinary levels of 5-HIAA, the most accurate biomarker of CS. However, in the majority of the relevant studies, the biochemical response appears at a lesser extent than the clinical response” (lines 94-98).
- Any data regarding the effect of high-dose/uncoventional SSAs on symptoms control?
As suggested by the reviewer, data on the effect of high dose SSAs are added in the revised manuscript.
“The first treatment options in refractory CS are either a dose escalation of SSAs, or the shortening of the interval between SSA administration or the additional administration of short-acting SSAs. Three systematic reviews have concluded that high-dose SSAs are well-tolerated and achieve significant rates of disease control. In patients with carcinoid symptoms refractory to octreotide LAR or lanreotide AG, pasireotide LAR showed non-inferiority in symptom control in comparison to high dose (40 mg) octreotide LAR.” (lines 108-113).
“Dose escalation of SSAs to above-label doses in refractory CS, although still unapproved, has proven efficacy in symptom relief in several studies and is encouraged by the present guidelines, taking into account the AE profile of these drugs. Two phase III studies investigated octreotide LAR in doses of 40 and 60 mg every 28 days and reported that, while these dosages are effective in controlling CS symptoms they were not associated with significant AE profile, with cholelithiasis and hyperglycemia being the most frequent events, occurring in all cases in less than 10% of patients. Similarly, a phase II study reported the AE profile of lanreotide AG 120 mg every 14 days, that mainly included gastrointestinal disorders (two patients with hyperglycemia and single cases with cholelithiasis and cholecystitis).” (lines 434-443).
- Please carefully double check the paper for some grammar (i.e. this..provides, please add s, see discussion first line) and typo mistakes
We would like to thank the reviewer for his remark and double-checked the text for grammar/typo mistakes.
Reviewer 2 Report
The authors have conducted a syst rev and meta-anlyzes of published studies of SSA to evaluate symptomatic response.
I have some concerns:
The main one is the meta-analyzes per se. The studies are too heterogeneous in terms of SSA dose and setting (some 1L, some in refractory carcinoid syndrome). So I am not sure it is scientifically sound to pool these studies together. I think a pure systematic review would be more adequate.
Unless, you could pool studies outcomes by line of therapy and SSA dosage.
NETTER-1 reports symptom control (within QoL evaluation) for the SSA-higher dose arm.
Minor:
line 95-96: please correct as SSA (or any other therapy) has led to OS gain in NET.
Author Response
Review Report #2
The authors have conducted a syst rev and meta-anlyzes of published studies of SSA to evaluate symptomatic response.
I have some concerns:
The main one is the meta-analyzes per se. The studies are too heterogeneous in terms of SSA dose and setting (some 1L, some in refractory carcinoid syndrome). So I am not sure it is scientifically sound to pool these studies together. I think a pure systematic review would be more adequate.
Unless, you could pool studies outcomes by line of therapy and SSA dosage.
We fully understand the reviewer’s concern. Due to the large heterogeneity documented, we separately present the peculiarities of individual studies in the part of the systematic review. This part is now significantly expanded in the revised version of the manuscript (Lines 230-311).
Pooling of the studies outcomes is not feasible due to the rather small number of included studies and individuals included there. However, even when considering this heterogeneity, the meta-analysis points out the effectiveness of SSAs in the control of diarrhea and flushing and we believe this finding is relevant for decision making in the management of CS. Still, the heterogeneity is acknowledged and pointed out in the limitations of the study and the need for further prospective studies is stressed out in the manuscript (lines 370-379, 459-463).
NETTER-1 reports symptom control (within QoL evaluation) for the SSA-higher dose arm.
As indicated by the reviewer, this statement is clarified in the revised version of the manuscript (lines 125-129).
Minor:
line 95-96: please correct as SSA (or any other therapy) has led to OS gain in NET.
As indicated by both reviewers, this misleading statement has been removed from the revised version of the manuscript.
Round 2
Reviewer 1 Report
The authors have addressed the issues
Reviewer 2 Report
Suggestions properly addressed.